# Kidney stone disease increases the risk of cardiovascular events

**Yuxuan Chen[1], XueWen Liao** [2]*

**1** Shengli Clinical Medical College, Fujian Medical University, Fuzhou, Fujian, China, **2** Department of Cardiology, Fuzhou University Affiliated Provincial Hospital, Fujian Provincial Hospital, Fuzhou, Fujian, China

* XuewenLiao@fzu.edu.cn

## Abstract

### Introduction

Kidney stone disease is associated with numerous cardiovascular risk factors. However, the findings across studies are non-uniformly consistent, and the control of confounding variables remains suboptimal. This study aimed to investigate the association between kidney stone and cardiovascular disease.

### Methods

We conducted an observational study using data from the National Health and Nutrition Examination Survey conducted between 2007 and 2010. Weighted multivariable-adjusted logistic regression was used to evaluate the association between kidney stones and cardiovascular event risk. Moreover, in observational studies, Mendelian randomization (MR) was applied to avoid reverse causality and reduce the influence of potential confounding factors. Inverse-variance weighted (IVW) was the main analytical method.

### Results

After controlling for cardiovascular and kidney stone risk factors among 7210 US adults, along with other potential confounding variables, patients with kidney stones exhibited a significantly elevated risk of acute myocardial infarction (AMI) (odds ratio [OR], 1.88 [95% confidence interval [CI], 1.09–3.26], $P < 0.05$). However, a non-significant association was observed with heart failure, hypertension, or stroke. MR analyses further indicated that genetically predicted kidney stones were causally associated with an increased risk of coronary heart disease (OR, 1.07 [95% CI, 1.04–1.53], $P = 0.028$), myocardial infarction (OR, 1.08 [95% CI, 1.02–1.15], $P = 0.015$), hypertension (OR 1.01 [95% CI, 1.00–1.02], $P = 0.042$) and ischemic stroke (OR, 0.86 [95% CI, 0.75–0.98], $P = 0.022$) in IVW models, with non-significant associations detected for heart failure.

**Data availability statement:** The data of this study are publicly available for free from the NHANES database (https://www.cdc.gov/nchs/nhanes/).

**Funding:** The author(s) received no specific funding for this work.

**Competing interests:** The authors have declared that no competing interests exist.

## Conclusions

The occurrence of kidney stones has been associated with an elevated risk of myocardial infarction within the context of cardiovascular events. However, cross-sectional analyses yield results that are inconsistent with those obtained from Mendelian randomization analyses regarding outcomes such as heart failure, hypertension, and stroke.

## 1 Introduction

Kidney stones represent a prevalent medical condition characterized by the crystallization of urinary solutes into aggregates within the urinary system [1], with their incidence showing an upward trend [2]. The current worldwide prevalence of nephrolithiasis is estimated to range between 7.2% and 7.7% [3]. The incidence of kidney stones seems to be gender specific. A recent estimate from the National Health and Nutrition Examination Survey (NHANES), a comprehensive dataset representing the US population, indicated that approximately one in ten adult males in the United States suffers from kidney stones. Conversely, this condition was less common among females, affecting around 7% of them [4]. Studies have demonstrated that the development of nephrolithiasis is affected by genetic predispositions, environmental conditions, dietary habits, physical activity levels, and various other determinants [5]. Kidney stones are linked to systemic conditions, including hypertension, diabetes, and dyslipidemia, which are recognized as risk factors for cardiovascular disease (CVD) [2,6–8]. Consequently, it is imperative to elucidate the relationship between kidney stones and cardiovascular disease. Several studies suggested that patients with kidney stones are at an increased risk of developing adverse health outcomes, such as chronic kidney disease and CVD. However, epidemiological research has yielded inconsistent findings regarding the association between kidney stones and cardiovascular risk. Some studies have indicated a positive correlation [6], while others have not been able to establish a substantial connection [9]. Published studies often fail to consider various potential confounding variables [2,4,10,11]. The observed association between kidney stones and cardiovascular events might be attributed to the indirect influences of shared risk factors [6].

Given the rising incidence of kidney stones over the years, a precise evaluation of their associated cardiovascular risks is essential for alerting patients and facilitating effective health interventions. Patients with kidney stones should focus on modifying cardiovascular risk factors to mitigate the comorbidity risk of both conditions. Furthermore, elucidating the relationship between KSD and cardiovascular events is of substantial importance for advancing our understanding of the etiology of these diseases.

We conducted a cross-sectional study based on NHANES to determine the relationship between kidney stones and cardiovascular events. Following this, a two-sample Mendelian randomization (MR) was applied to further explore the causal relationship between these two conditions, employing genetic markers associated

with kidney stones as instrumental variables to infer causal relationships with specific cardiovascular disease outcomes. This approach is more resistant to biases associated with confounding variables and the issue of reverse causality that can plague traditional observational research [12].

## 2 Methods

### 2.1 Study population in NHANES

The NHANES, an ongoing two-year cycle nationally representative survey, is an essential research initiative focused on assessing the health and nutritional conditions of the U.S. population, including adults and children. The NHANES protocols were approved by the Research Ethics Review Board of the National Center for Health Statistics, and written informed consent was obtained from all participants. In this study, we downloaded the NHANES data from 2007 to 2010, as these two cycles included information on kidney stones and CVD.

This study selected two NHANES cycles from 2007 to 2010 to assess the association between kidney stones and cardiovascular events. A total of 20,688 participants were initially enrolled. After excluding participants aged < 18 years (n = 7933), those with incomplete data on KSD (n = 645) or cardiovascular events (n = 61), those with incomplete data on other covariates (n = 4767), and those with recorded estimated glomerular filtration rate (eGFR) values under 15 ml/min/1.73 m$^2$, as well as those who had undergone dialysis or kidney transplantation (n = 72). 7210 participants were included in our final analysis (Fig 1).

### 2.2 Variables included in NHANES

During individual interviews, the diagnosis of kidney stones was ascertained using the Kidney Conditions-Urology survey within the questionnaire data. This survey was administered by trained interviewers in participants' homes, employing a computer-assisted personal interview (CAPI) system. Participants were queried with the question, "Have you ever had a kidney stone?" An affirmative response was used to classify the participant as having a history of kidney stones.

The outcome variables of this study are cardiovascular events, including myocardial infarction, congestive heart failure, hypertension, and stroke. These varibles were also obtained from the standardized healthcare status questionnaire in NHANES 2007–2010. The participants were also asked, " Has a doctor or other health expert ever informed you that you have myocardial infarction (MI)/congestive heart failure(CHF)/high blood pressure(HBP)/stroke?" An affirmative response to any of these questions qualified an individual as having CVD. Myocardial infarction, congestive heart failure, hypertension, and stroke are also defined according to the problems of the corresponding diseases mentioned above.

Information on different demographic and health-related factors was obtained from the NHANES household interviews, including age, sex, race/ethnicity, education level, C-reactive protein (CRP) levels, smoking status, disease conditions (diabetes, hyperlipidemia, and gout), and dietary intake (total calorie intake, calcium consumption, sodium intake, and total water intake). The body mass index (BMI) was determined by dividing an individual's weight (in kilograms) by the square of their height (in meters). BMI was categorized as underweight (< 18.5), normal weight (18.5–24.9), overweight (25.0–29.9), or obese (≥ 30.0). Race/ethnicity was categorized as Mexican American, other Hispanic, non-Hispanic White, non-Hispanic Black, or other races, including multi-racial, while education level was classified as less than 9th grade, 9–11th grade, high school graduate/GED, some college or AA degree, or college graduate or above. Smoking status was recorded as having smoked 100 cigarettes or more in one's lifetime. CRP status was divided into ≤ 1.0 or > 1.0. The eGFR was calculated utilizing the Chronic Kidney Disease Epidemiology Collaboration (CKD-EPI) equation. The eGFR values were stratified into the following categories: ≥ 60, 45–59.9, 30–44.9, and 15–29.9 mL/min/1.73 m$^2$.

### 2.3 Genetic instruments for KSD in MR

We identified genetic variations associated with KSD in the European population from the FinnGen Biobank. Researchers used ICD-10 codes to identify individuals with a history of kidney stones (S1 Table). Our genome-wide association study

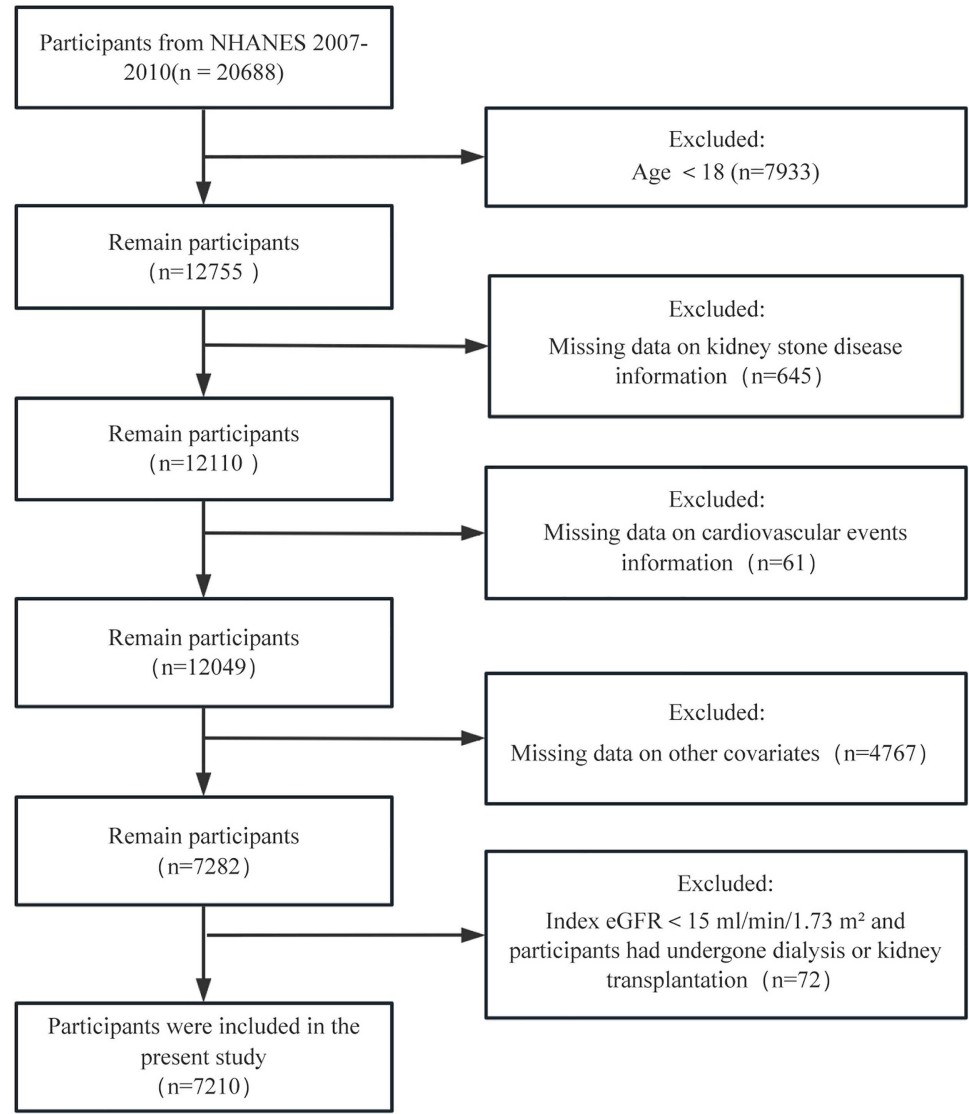

**Fig 1. Flowchart of study participants from NHANES 2007–2010.**

involved 218,792 individuals of European ancestry, including 5,347 cases and 213,445 controls. With a significance threshold of $P<5\times10^{-8}$, we discovered nine single-nucleotide polymorphisms (SNPs) that correlated with KSD. The robustness of these SNPs was assessed by calculating the F-statistics, which were found to be ≥ 10 for all identified KSD variants. The latest research suggests that sample overlap may lead to bias in causal estimation in MR analysis [13,14]. To minimize bias caused by sample overlap, MR analysis of outcomes from the FinnGen cohort should be avoided as much as possible when the exposure is concerned. When the outcomes are obtained from different cohorts, a meta-analysis should be performed to evaluate the overall effect. The study design of two-sample MR analysis was exhibited in Fig 2.

## 2.4 Genetic summary data for cardiovascular events

For CVDs, we detected five phenotypes, including coronary heart disease, MI, heart failure, hypertension, and stroke. Coronary heart disease (n = 184,305) and MI (n = 171,875) data were obtained from the CARDIoGRAMplusC4D

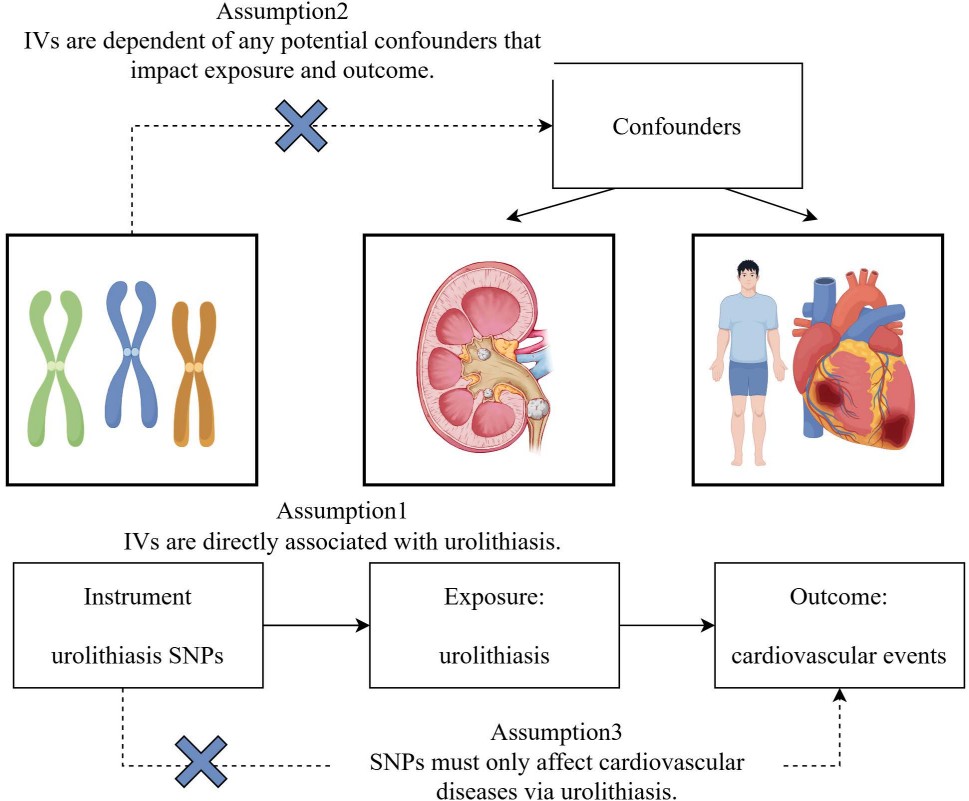

Assumption2
IVs are dependent of any potential confounders that impact exposure and outcome.

Confounders

Assumption1
IVs are directly associated with urolithiasis.

| Instrument | Exposure: | Outcome: |
|---|---|---|
| urolithiasis SNPs | urolithiasis | cardiovascular events |

Assumption3
SNPs must only affect cardiovascular diseases via urolithiasis.

**Fig 2. The study design of two-sample MR analysis.**

Consortium. Hypertension (n = 463,010) and heart failure (n = 361,194) data were obtained from the UK Biobank Consortium. Ischemic stroke (n = 977,323) data were obtained from the International Stroke Genomics Consortium.

## 2.5 Statistical analysis

To analyze the NHANES data, which necessitates a complex sampling design, we incorporated sample weights, clustering, and stratification in our analyses. Utilizing R software (version 4.2.1), available from the R Project for Statistical Computing (https://www.r-project.org), we employed the NHANES-recommended sample weights to amalgamate data from 2007 to 2008 and 2009–2010 two-year survey periods. This was achieved using a unique respondent sequence number.

Participants were categorized into two groups based on whether they had KSD or not. For continuous variables, we calculated the mean and standard deviation (SD), while categorical variables are presented as frequency and percentage. We then compared the baseline characteristics of the participants using a one-way analysis of variance for continuous variables and Pearson's chi-square test for categorical variables. To assess the association between depression and the risk of kidney stones, we performed multivariate-adjusted logistic regression. This analysis was stratified based on various factors, including age, gender, race, education level, BMI category, smoked 100 cigarettes, eGFR, diabetes, gout, and hyperlipidemia. The adjusted odds ratios (ORs), along with their 95% confidence intervals (CIs), were determined. Statistical significance was set at a $P$-value threshold of less than 0.05. All outcomes were evaluated using a two-tailed test.

An inverse-variance weighted (IVW) meta-analysis under a random-effects model was regarded as the primary analysis. The following two methods, weighted median and MR-Egger, were used for sensitivity analyses. MR-Egger method can be used to assess the horizontal pleiotropy of selected instrumental variables (IVs) [15]. Cochrane's Q statistic was

used to assess the variability among the chosen IVs. Moreover, a sensitivity analysis excluding one variable at a time was performed to evaluate whether the aggregate estimates were significantly influenced by any single SNP.

## 3 Results

### 3.1 Observational results between KSD and cardiovascular events in NHANES

**3.1.1 Characteristics of study participants.** In our observational study, we selected 7,210 individuals. Table 1 presents the baseline characteristics of the participants according to KSD status. We compared the demographic differences between participants with and without kidney stones (Table 1). Participants with kidney stones were significantly more likely to be older (55.7 versus 50.6 years), have a higher incidence of diabetes mellitus (17.5% versus 9.3%), a higher risk of gout (9.8% versus 4.1%), and a higher risk of hypercholesterolemia (51.3% versus 41.1%) (all $P<0.001$).

**3.1.2 Association between KSD and cardiovascular events.** We investigated the association between KSD and cardiovascular risks using logistic regression models (Table 2). In model 1, the risk of cardiovascular events among patients with a kidney stone history was higher compared to those without a kidney stone history. Compared with non-kidney stone individuals, ORs (95% CI) for risks of MI, heart failure, hypertension, and stroke among patients with kidney stones were 2.75 [95% CI, 1.93–3.93], 2.05 [95% CI, 1.48–2.83], 1.89 [95% CI, 1.50–2.38], and 1.92 [95% CI, 1.33–2.75], respectively. In model 2, the association between kidney stones and cardiovascular events risk remained significant. In models 3 and 4, kidney stone was only associated with an increased risk of AMI, but not heart failure, hypertension, or stroke. In model 4, significant associations were also observed between the development of kidney stones and the composite outcome of the individual events (OR, 1.34 [95% CI, 1.10–1.63]). The corresponding results are additionally provided in S1 Fig.

**3.1.3 Stratified analyses.** We conducted a subgroup analysis stratified by age, sex, race, educational attainment, BMI, smoked 100 cigarettes, eGFR, diabetes, gout and hypercholesterolemia to further explore the relationship between kidney stones and MI outcomes in different populations. The effect of kidney stones on the main outcome was consistent across the preassigned subgroups of gender, age, race, educational attainment, BMI, smoked 100 cigarettes, eGFR, diabetes, gout and hypercholesterolemia. Interaction tests indicated that the association between KSD and MI did not exhibit statistically significant differences across the various strata (Fig 3).

### 3.2 Causal association between KSD and cardiovascular events in MR

After selection, nine SNPs were used in the MR analysis. Detailed information on these SNPs is presented in S1 Table. Results revealed that genetically predicted KSD was causally correlated with an elevated risk of coronary heart disease (OR 1.07 [95% CI, 1.04–1.53], $P=0.028$), MI (OR, 1.08 [95% CI, 1.02–1.15], $P=0.015$), and hypertention (OR 1.01 [95% CI, 1.00–1.02], $P=0.042$) in IVW. Surprisingly, the occurrence of kidney stones reduced the risk of ischemic stroke (OR 0.86 [95% CI, 0.75–0.98], $P=0.022$). However, a non-significant correlation was found in heart failure (OR 1.00 [95% CI, 0.99–1.00], $P=0.904$) (Table 3), and a non-significant correlation was found in the MR-Egger. There was no pleiotropy in any of the cardiovascular event analyses, and some outcomes exhibited heterogeneity. Scatter plot and forest plot illustrating the association between KSD and cardiovascular events are presented in S2 and S3 Figs, respectively, where similar results can be observed.

## 4 Discussion

Our ongoing research has identified a correlation between the occurrence of kidney stones and specific CVD. The consistency between the outcomes of observational studies and MR analyses reinforces the reliability of this discovery.

In a seminal study conducted by Elmfeldt [16] in 1976, a correlation between kidney stones and CVD was first established. However, the study population was limited to males and only adjusted for age, and there were no other studies available at the time to confirm these results.

**Table 1. Baseline characteristics of participants categorized by kidney stone disease status in the NHANES 2007–2010 study.**

| Variable | Overall | No Stone | Stone | P-value |
|---|---|---|---|---|
| Sex, n (%) | | | | <0.001 |
| Male | 3366 (45.5) | 2893 (44.0) | 473 (57.4) | |
| Female | 3844 (54.5) | 3529 (56.0) | 315 (42.6) | |
| Age (years) | 51.2 (15.5) | 50.6 (15.6) | 55.7 (14.4) | <0.001 |
| Race, n (%) | | | | <0.001 |
| Mexican American | 1047 (5.9) | 960 (6.1) | 87 (4.2) | |
| Other Hispanic | 722 (4.1) | 641 (4.0) | 81 (4.1) | |
| Non-Hispanic White | 3807 (75.0) | 3301 (74.1) | 506 (82.4) | |
| Non-Hispanic Black | 1336 (10.0) | 1245 (10.6) | 91 (5.4) | |
| Others | 298 (5.0) | 275 (5.2) | 23 (3.9) | |
| Education attainment, n (%) | | | | 0.017 |
| Less Than 9th Grade | 793 (5.0) | 695 (4.9) | 98 (6.3) | |
| 9-11th Grade | 1042 (10.9) | 923 (10.8) | 119 (11.1) | |
| High School Grad/GED | 1599 (22.0) | 1402 (21.6) | 197 (25.3) | |
| Some College or AA degree | 2043 (30.6) | 1822 (30.4) | 221 (32.6) | |
| College Graduate or above | 1733 (31.5) | 1580 (32.3) | 153 (24.7) | |
| BMI category, n (%) | | | | 0.011 |
| Underweight | 78 (1.2) | 74 (1.3) | 4 (0.9) | |
| Normal weight | 1705 (26.1) | 1568 (27.0) | 137 (17.8) | |
| Overweight | 2483 (34.2) | 2211 (34.0) | 272 (35.7) | |
| Obesity | 2944 (38.5) | 2569 (37.7) | 375 (45.6) | |
| Smoked 100 cigarettes, n (%) | | | | 0.14 |
| Yes | 3325 (45.0) | 2929 (44.6) | 396 (48.2) | |
| No | 3885 (55.0) | 3493 (55.4) | 392 (51.8) | |
| Intake | | | | |
| Total calories, kcal | 2056.2(803.0) | 2,047.5(798.4) | 2,128.8(837.2) | 0.2 |
| Calcium, mg | 967.0(505.2) | 971.4(512.9) | 929.9(434.7) | 0.4 |
| Sodium, mg | 3,455.1(1,514.8) | 3,439.6(1,492.5) | 3,584.4(1,684.7) | 0.2 |
| Total plain water drank, g | 993.9(937.3) | 1,001.1(936.8) | 933.7(940.2) | 0.047 |
| CRP status, n (%) | | | | 0.6 |
| Low | 6463 (91.0) | 5759 (91.1) | 704 (90.5) | |
| High | 747 (9.0) | 663 (8.9) | 84 (9.5) | |
| eGFR, n (%) | | | | 0.4 |
| 15–29.9 mL/min/1.73m$^2$ | 44 (0.4) | 39 (0.4) | 5 (0.7) | |
| 30–44.9 mL/min/1.73m$^2$ | 154 (1.2) | 130 (1.2) | 24 (1.6) | |
| 45–59.9 mL/min/1.73m$^2$ | 373 (3.0) | 319 (3.0) | 54 (3.5) | |
| ≥60 mL/min/1.73m$^2$ | 6639 (95.2) | 5934 (95.4) | 705 (94.2) | |
| Diabetes, n (%) | | | | <0.001 |
| Yes | 1081 (10.2) | 897 (9.3) | 184 (17.5) | |
| No | 6129 (89.8) | 5525 (90.7) | 604 (82.5) | |
| Myocardial infarction, n (%) | | | | <0.001 |
| Yes | 394 (3.9) | 310 (3.3) | 84 (8.7) | |
| No | 6816 (96.1) | 6112 (96.7) | 704 (91.3) | |
| Heart failure, n (%) | | | | <0.001 |
| Yes | 270 (2.6) | 222 (2.4) | 48 (4.7) | |
| No | 6940 (97.4) | 6200 (97.6) | 740 (95.3) | |

*(Continued)*

**Table 1.** (Continued)

| Variable | Overall | No Stone | Stone | P-value |
|---|---|---|---|---|
| Hypertension, n (%) | | | | <0.001 |
| Yes | 3149 (37.1) | 2709 (35.5) | 440 (51.0) | |
| No | 4061 (62.9) | 3713 (64.5) | 348 (49.0) | |
| Stroke, n (%) | | | | <0.001 |
| Yes | 334 (3.4) | 271 (3.1) | 63 (5.8) | |
| No | 6876 (96.6) | 6151 (96.9) | 725 (94.2) | |
| Gout, n (%) | | | | <0.001 |
| Yes | 420 (4.7) | 329 (4.1) | 91 (9.8) | |
| No | 6790 (95.3) | 6093 (95.9) | 697 (90.2) | |
| Hypercholesterolemia, n (%) | | | | 0.001 |
| Yes | 3229 (42.2) | 2810 (41.1) | 419 (51.3) | |
| No | 3981 (57.8) | 3612 (58.9) | 369 (48.7) | |

For continuous variables, the p-value was determined using a weighted one-way analysis of variance, while for categorical variables, it was calculated using a weighted chi-square test.

**Table 2.** The association between kidney stone disease and cardiovascular events.

| Models | Cardiovascular Events | | | | |
|---|---|---|---|---|---|
| | Myocardial infarction | Heart failure | Hypertension | Stroke | All CV events |
| | OR (95%CI) | OR (95%CI) | OR (95%CI) | OR (95%CI) | OR (95%CI) |
| Model1 | 2.75 (1.93, 3.93) *** | 2.05 (1.48, 2.83) *** | 1.89 (1.50, 2.38) *** | 1.92 (1.33, 2.75) *** | 2.07(1.77,2.41) *** |
| Model2 | 1.97 (1.34, 2.90) ** | 1.51 (1.04, 2.19) * | 1.57 (1.23, 2.02) *** | 1.57 (1.08, 2.29) * | 1.62(1.38,1.91) *** |
| Model3 | 1.83 (1.15, 2.94) * | 1.36 (0.90, 2.06) | 1.30 (0.99, 1.70) | 1.47 (0.99, 2.19) | 1.42(1.18,1.70) *** |
| Model4 | 1.88(1.09, 3.26) * | 1.15 (0.72,1.85) | 1.27 (0.95, 1.69) | 1.39 (0.91, 2.13) | 1.34(1.10,1.63) *** |

OR: odds ratio. 95% CI: 95% confidence interval.

* $p < 0.05$.

** $p < 0.01$.

*** $p < 0.001$.

Model 1: crude model. Model 2: adjusted for demographic characteristics including age, gender, race and education attainment. Model 3: adjusted for age, gender, race, education attainment, BMI category, smoked 100 cigarettes, total calories, calcium intake, CRP status, eGFR, diabetes, gout and hypercholesterolemia. Model 4: adjusted for age, gender, race, education attainment, BMI category, smoked 100 cigarettes, total calories, calcium intake, CRP status, eGFR, diabetes, gout, hypercholesterolemia and other CVDs.

Recently, Ferraro and Taylor [2] published the results of three large prospective cohort studies, including the Health Professionals Follow-up Study (HPFS), Nurses' Health Study I (NHS I), and Nurses' Health Study II (NHS II). The findings of this study align with those of our research. After adjusting for potential confounding factors, the hazard ratio (HR) for developing coronary heart disease in patients with stones in the NHS I cohort was 1.18 [95% CI, 1.08–1.28], and in the NHS II cohort, it was 1.48 [95% CI, 1.23–1.78]; however, this correlation disappeared in the HPFS cohort. This study found a gender difference between a history of kidney stones and the risk of coronary heart disease, although many studies have reported this phenomenon. This may be because women are more likely than men to be exposed to potential factors that increase the risk of cardiovascular and kidney stones. Although the authors investigated the composite outcomes of coronary heart disease, they were limited to MI. Our cardiovascular events included MI, heart failure, hypertension, and stroke. The article also lacks laboratory data, such as serum creatinine information, to exclude the potential impact of renal function on CVD.

| Subgroup | OR (95% CI) | P-value | P for interaction |
|---|---|---|---|
| **Age** | | | 0.308 |
| 18-49.9 | 1.02 (1.00, 1.04) | 0.100 | |
| 50-69.9 | 1.05 (1.00, 1.10) | 0.027 | |
| ≥70 | 1.03 (0.97, 1.11) | 0.300 | |
| **Gender** | | | 0.663 |
| Male | 1.05 (1.00, 1.09) | 0.049 | |
| Female | 1.01 (0.98, 1.05) | 0.400 | |
| **Race** | | | 0.630 |
| Mexican American | 1.04 (0.96, 1.13) | 0.300 | |
| Other Hispanic | 1.00 (0.95, 1.06) | >0.900 | |
| Non-Hispanic White | 1.04 (1.01, 1.07) | 0.017 | |
| Non-Hispanic Black | 0.99 (0.94, 1.04) | 0.400 | |
| Other Race-Including Multi-Racial | 1.00 (0.87, 1.14) | >0.900 | |
| **Educational attainment** | | | 0.900 |
| Less Than 9th Grade | 1.09 (0.99, 1.21) | 0.081 | |
| 9-11th Grade | 1.03 (0.94, 1.12) | 0.500 | |
| High School Grad/GED | 1.03 (0.98, 1.08) | 0.200 | |
| Some College or AA degree | 1.03 (1.00, 1.07) | 0.070 | |
| College Graduate or above | 1.03 (0.96, 1.10) | 0.400 | |
| **BMI category** | | | 0.320 |
| Underweight | 0.99 (0.96, 1.03) | 0.300 | |
| Normal weight | 1.01 (0.97, 1.04) | 0.700 | |
| Overweight | 1.02 (0.97, 1.07) | 0.500 | |
| Obesity | 1.06 (1.02, 1.09) | 0.003 | |
| **Smoked 100 cigarettes** | | | 0.820 |
| Yes | 1.05 (1.00, 1.10) | 0.066 | |
| No | 1.02 (0.99, 1.06) | 0.200 | |
| **eGFR** | | | 0.984 |
| 15-29.9 mL/min/1.73m$^2$ | 1.06 (0.57, 1.95) | 0.800 | |
| 30-44.9 mL/min/1.73m$^2$ | 1.24 (0.95, 1.62) | 0.110 | |
| 45-59.9ml/min/1.73m$^2$ | 1.07 (0.90, 1.26) | 0.400 | |
| ≥60 mL/min/1.73m$^2$ | 1.03 (1.00, 1.06) | 0.046 | |
| **Diabetes** | | | 0.870 |
| Yes | 1.06 (0.98, 1.14) | 0.110 | |
| No | 1.03 (1.00, 1.06) | 0.064 | |
| **Gout** | | | 0.071 |
| Yes | 0.98 (0.89, 1.08) | 0.700 | |
| No | 1.04 (1.01, 1.07) | 0.013 | |
| **Hypercholesterolemia** | | | 0.529 |
| Yes | 1.04 (1.00, 1.09) | 0.049 | |
| No | 1.02 (0.99, 1.06) | 0.110 | |

0.50    1.00    1.50    2.00

**Fig 3. Forest plots of AMI events in NHANES 2007–2010, by strata.** The plots exhibit the odd adjusted ratios (with corresponding 95% CIs) of AMI associated with kidney stone presentation during the study. *P* for interaction are a measure of the interaction between each characteristic and the risk of AMI associated with the KSD. Odds ratios were adjusted for age, sex, race, education attainment, BMI category, smoking 100 cigarettes, total calories, calcium intake, CRP status, diabetes, gout, hypertension and hypercholesterolemia.

Chien Yi Hsu [17] also published a longitudinal study on the association between urinary tract stones and the risk of MI and stroke in a population-based cohort database in Taiwan Province, China. After 10 years of follow-up, patients with urinary tract stones had an increased risk of future MI (HR, 1.31 [95% CI, 1.09–1.56]), stroke (HR, 1.39 [95% CI, 1.24–1.55], and total cardiovascular events (HR, 1.38 [95% CI, 1.25–1.51]) than those in the control group. The study additionally

**Table 3. Mendelian randomization estimates for the association between kidney stone disease and cardiovascular events.**

| Outcome | Inverse variance weighted | | MR-Egger | | Pleiotropy | | Heterogeneity | |
|---|---|---|---|---|---|---|---|---|
| | OR (95% CI) | p-value | OR (95% CI) | p-value | Intercept | p-value | Q | p-value |
| Coronary heart disease | 1.07 (1.04, 1.53) | 0.028 | 1.23 (0.80, 1.89) | 0.381 | −0.0198 | 0.554 | 11.56 | 0.172 |
| Myocardial infarction | 1.08 (1.02, 1.15) | 0.015 | 1.58 (1.07, 2.35) | 0.062 | −0.0551 | 0.104 | 7.10 | 0.418 |
| Heart failure | 1.00 (0.99, 1.00) | 0.904 | 1.00 (0.99, 1.01) | 0.405 | −0.0001 | 0.392 | 15.98 | 0.030 |
| Hypertension | 1.01 (1.00, 1.02) | 0.042 | 1.01 (0.96, 1.07) | 0.715 | −0.0009 | 0.834 | 31.47 | <0.001 |
| Stroke | 0.86 (0.75, 0.98) | 0.022 | 0.46 (0.21, 1.01) | 0.101 | 0.09031 | 0.166 | 8.77 | 0.269 |
| All CV events | 1.01 (0.99, 1.02) | 0.307 | 1.02 (0.95, 1.09) | 0.640 | – | – | – | – |

OR, odds ratio; CI, confidence interval; IVW, inverse variance weighted; MRPRESSO, Mendelian randomization-pleiotropy residual sum and outlier.

incorporated variables such as the location of stones and the types of stone surgery, which was unavailable in previous studies.

The potential explanation of the relationship between kidney stones and cardiovascular events mostly focuses on the mediating effect, one of which is that kidney stones increase the incidence rate of certain cardiovascular risk factors (such as diabetes [18], hypertension [19,20], metabolic syndrome [21,22], and other diseases) and indirectly increase the risk of CVD. Alternatively, due to certain dietary or medication factors, kidney stones are associated with cardiovascular disease. For example, calcium intake (including dietary sources and supplements) may affect the risk of kidney stones and hypertension [23,24]. Thiazide drugs reduce the risk of kidney stones and hypertension and coronary heart disease [25,26]. However, the latest research exhibits that there is a non-significant difference in recurrence rate between patients with recurrent kidney stones who receive any dose of hydrochlorothiazide treatment and those who receive a placebo [27]. Moreover, kidney stones can contribute to the development of chronic kidney disease (CKD) by impairing renal function [28], and there is no doubt that CKD leads to an increase in the incidence rates and mortality of CVD [29,30].

Besides clinical research, basic experimental studies have also provided possible mechanisms for the potential pathways from stone formation to CVD. Chronic low systemic inflammation and oxidative stress may lead to kidney stone formation and cardiovascular events [31]. Oxidative stress and inflammatory state jointly promote the occurrence of endothelial dysfunction (ED), which can cause the imbalance of oxidative and antioxidant balance reactions in cells, leading to a cascade reaction of signaling pathways and an increase in inflammatory markers such as NADPH oxidase, adhesion molecules, COX-2, tumor necrosis factor α, interleukin-6, CRP, 8-hydroxydeoxyguanosine, 3-nitrotyrosine, and monocyte chemoattractant proteins [32–34]. This leads to inflammation and fibrosis, which damages kidney function. Our study found that individuals without a baseline history of hypertension tended to have an increased risk of AMI associated with kidney stones. Perhaps due to the lack of competitive factors for cardiovascular events, such as hypertension, the impact of hypertension is relatively small, and the risk of AMI is relatively increased. This amplifies the risk of cardiovascular events; however, this explanation is speculative. Moreover, other traditional CVD-related factors, such as hyperlipidemia, smoking, or hypertension, are mechanistically associated with ED and oxidative stress [35]. This may be another possible reason why the risk of kidney stone-related AMI still exists, even after adjusting for relevant CVD risk factors.

Our findings indicate a significant association between kidney stones and an elevated risk of MI. After controlling for multiple variables, a non-significant association was found between kidney stones and other cardiovascular outcomes. One plausible explanation for this discrepancy is that individuals with kidney stones often undergo medical or surgical interventions, which may mitigate the severity and associated risks of other cardiovascular events.

In our study, we observed a non-significant gender-stratified association between a history of kidney stones and the risk of coronary heart disease. This finding contrasts with numerous previous meta-analyses [36,37] that have consistently

reported gender disparities in the relationship between kidney stones and cardiovascular events. Recent research has highlighted gender-specific differences in microbial diversity as potential contributors to the varying risks of kidney stones that can precipitate coronary heart disease [38]. Furthermore, several studies have implicated racial disparities in this context. A study by Glover demonstrated that kidney stones are linked to a heightened 10-year risk of future atherosclerotic cardiovascular disease events in non-Hispanic Black populations [39]. These differences may be due to the influence of unique and unknown factors in ethnic groups, increasing the risk of CVD and kidney stones.

In light of the observed correlation, it is advisable to implement screening protocols for cardiac dysfunction in patients with kidney stones. Patients diagnosed with urolithiasis may have an additional risk factor for CVD. This finding introduces novel perspectives for incorporating kidney stones into the risk stratification management of cardiovascular diseases. Evidence indicates that implementing lifestyle modifications may contribute to the prevention of both conditions [40,41]. By initiating early lifestyle interventions, such as DASH diet and exercise for weight reduction, alongside established risk factor prevention and control strategies, namely, the management of blood pressure, blood glucose, and lipid levels, this comprehensive approach may contribute to a reduction in the incidence of cardiovascular events among patients with urolithiasis [42]. Future prospective studies should be structured to determine whether interventions targeting kidney stones might mitigate the healthcare costs associated with CVDs, thereby potentially decreasing the need for unnecessary diagnostic procedures and pharmacological treatments in patients over the long term.

The advantage of this study lies in its comprehensive analysis of nationally representative samples and careful consideration of multiple confounding factors. Our study has several limitations that warrant consideration. The reliance on self-reported kidney stone history in the NHANES introduces potential biases, as such data are subject to underreporting and inaccuracies. Furthermore, the absence of detailed stone composition data in the NHANES and Finnish Biobank precludes the formulation of a comprehensive etiological hypothesis. Our analysis was further constrained by the exclusion of participants with incomplete data, which may have introduced a selection bias. However, we endeavored to mitigate this by employing sampling weights designed to adjust for non-response and ensure nationally representative estimates. The omission of sensitivity analysis in this study limits our ability to assess the robustness of the findings to different analytical assumptions. Finally, our results, derived from European and American populations, may not be generalizable to all ethnic groups, highlighting the need for further research in diverse populations.

## 5 Conclusion

In our study, which integrated NHANES data with MR analysis, we observed an elevated risk of certain CVD in individuals with a history of kidney stones. This association warrants further validation through rigorous studies. The underlying mechanisms linking kidney stones to CVD remain to be elucidated and necessitate more in-depth investigations.

## Supporting information

**S1 Table. Genetic instruments used in this MR study.**
(DOCX)

**S1 Fig. Multi -state results: ORs and 95% CIs of cardiovascular events.** Model 1 adjusted for none. Model 2 adjusted for age, gender, race and education attainment. Model 3 further adjusted for BMI category, smoked 100 cigarettes, total calories, calcium intake, CRP status, eGFR, diabetes, gout and hypercholesterolemia. Model 4 further adjusted for other CVDs beyond the primary outcome. Abbreviations: OR, Odds Ratio; CI, confidence interval; CVDs, cardiovascular diseases.
(TIF)

**S2 Fig. Scatter plots for MR analyses of the causal effect of kidney stone disease on cardiovascular events.** (A) Coronary heart disease. (B) Myocardial infarction. (C) Heart failure. (D) Hypertension. (E) Stroke.
(TIF)

**S3 Fig. Forest plot to visualize causal effect of each single SNP of kidney stone disease and cardiovascular events.** (A) Coronary heart disease. (B) Myocardial infarction. (C) Heart failure. (D) Hypertension. (E) Stroke.
(TIF)

## Acknowledgments

The authors sincerely thank the authors who shared the original dataset in this study.

## Author contributions

**Conceptualization:** Yuxuan Chen, XueWen Liao.

**Data curation:** Yuxuan Chen, XueWen Liao.

**Funding acquisition:** XueWen Liao.

**Investigation:** Yuxuan Chen.

**Methodology:** Yuxuan Chen.

**Project administration:** Yuxuan Chen, XueWen Liao.

**Software:** Yuxuan Chen.

**Supervision:** Yuxuan Chen, XueWen Liao.

**Validation:** Yuxuan Chen.

**Visualization:** Yuxuan Chen.

**Writing – original draft:** Yuxuan Chen.

**Writing – review & editing:** XueWen Liao.

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
