## [Decision Letter · Decision Letter 0]

8 Apr 2025

Dear Dr. Liao,

Thank you for submitting your manuscript to PLOS ONE. After careful consideration, we feel that it has merit but does not fully meet PLOS ONE’s publication criteria as it currently stands. Therefore, we invite you to submit a revised version of the manuscript that addresses the points raised during the review process.

Please submit your revised manuscript by May 23 2025 11:59PM. If you will need more time than this to complete your revisions, please reply to this message or contact the journal office at plosone@plos.org . A rebuttal letter that responds to each point raised by the academic editor and reviewer(s). You should upload this letter as a separate file labeled 'Response to Reviewers'.A marked-up copy of your manuscript that highlights changes made to the original version. You should upload this as a separate file labeled 'Revised Manuscript with Track Changes'.An unmarked version of your revised paper without tracked changes. You should upload this as a separate file labeled 'Manuscript'.

We look forward to receiving your revised manuscript.

Kind regards,

Ricardas Radisauskas

Academic Editor

PLOS ONE

Additional Editor Comments:

Dear manuscript authors,

Thank you for your submitted manuscript.

The manuscript still has some essential shortcomings, which the authors must correct based on the reviewers' comments.

Reviewers' comments:

Reviewer's Responses to Questions

**Comments to the Author**

1. Is the manuscript technically sound, and do the data support the conclusions?

Reviewer #1: Partly

Reviewer #2: Yes

Reviewer #3: Yes

2. Has the statistical analysis been performed appropriately and rigorously?

Reviewer #1: Yes

Reviewer #2: I Don't Know

Reviewer #3: Yes

3. Have the authors made all data underlying the findings in their manuscript fully available?

Reviewer #1: Yes

Reviewer #2: Yes

Reviewer #3: Yes

4. Is the manuscript presented in an intelligible fashion and written in standard English?

Reviewer #1: Yes

Reviewer #2: Yes

Reviewer #3: Yes

Reviewer #1: Its an interesting article showing association of Kidney stone and cardiovascular risk. I recommend below improvements -

- Prevalence - provide a number in your introduction

- diagnosis of kidney stone ? was it made by Imaging or for someone who required intervention - lithotripsy? please clarify

- biggest limitation is no known renal function - no creatinine values? which would be the biggest confounder. This is well established that renal stone will cause CKD and CKD is a major risk for cardiovascular disease. is it possible to obtain lab values ?

Reviewer #2: Given the topic of kidney stone data and evaluating associations, the data is presented in an extremely complex way. For an average reader of this article, it has dense amount of statistical analysis without clarity.

Table 1 also needs better explanation. Text states that 7377 individuals selected but N was 7282 (including the Forest plot). Secondly, what is the "N" in No or Yes? That has no explanation and these multiple "N" categories are confusing.

In section 3.1.2 authors state heart failure was not significant but In Table 2, Heart Failure appears to be statistically significant. There seems to be discrepancy.

Reviewer #3: The manuscript explores a relevant topic with valuable implications and is based on a sound methodological framework. However, improvements are needed in clarity, organization, data presentation, and language to strengthen its overall quality and impact.

Major comments

- The objectives of the study need to be stated more clearly in the introduction. Currently, the aims are somewhat vague and scattered.

- More details should be provided regarding the study design and participant selection. For instance, information about sample size calculation, inclusion/exclusion criteria,

- Tables and figures need to be formatted more clearly, with consistent use of labels, units, and statistical indicators (e.g., p-values, confidence intervals).Some of the figures are difficult to interpret due to lack of descriptive legends and clarity.

- The discussion would benefit from a deeper integration of the study’s findings with existing literature, as there is little critical comparison with prior research. The implications of the results and potential directions for future studies are also underdeveloped. Additionally, language issues—such as “did not explained” instead of “did not explain” and phrases like “consent of participation”—negatively impact readability. A comprehensive proofreading or professional language review is strongly recommended.

**Do you want your identity to be public for this peer review?** For information about this choice, including consent withdrawal, please see our Privacy Policy

Reviewer #1: No

Reviewer #2: No

Reviewer #3: No

---

## [Author Response · Author response to Decision Letter 1]

6 Jun 2025

Dear Editors and Reviewers:

Thank you for your letter and for the reviewers’ comments concerning our manuscript entitled “Kidney stone disease increases the risk of cardiovascular events” (ID: PONE-D-25-10263). Those comments are all valuable and very helpful for revising and improving our paper, as well as the important guiding significance to our researches. We have studied comments carefully and have made correction which we hope meet with approval.

To easily distinguish my answers from reviews’ comments, we highlight all of our answers in blue while keeping your letter and reviews’ comments in black in the Response letter.

Thank you again for your time and consideration.

Looking forward to hearing from you.

Referee Comments to Author

Reviewer #1:

Comments to the Author

1.Prevalence - provide a number in your introduction.

Reply: We gratefully appreciate for your valuable suggestion.

According to your comments, we have added a detailed description of the prevalence of kidney stones in the introduction section, and the revised sentence is presented as follows: “The worldwide prevalence of nephrolithiasis is estimated to range between 7.2% and 7.7%[1]” (pages 1-2).

2. Diagnosis of kidney stone? was it made by Imaging or for someone who required intervention - lithotripsy? please clarify.

Reply: We extend our sincere appreciation for the reviewer's insightful suggestions.

In the NHANES database, the diagnosis of kidney stones was ascertained using the Kidney Conditions-Urology survey within the questionnaire data. This survey was administered by trained interviewers in participants' homes, employing a computer-assisted personal interview (CAPI) system. Participants were queried with the question, "Have you ever had a kidney stone?" An affirmative response was used to classify the participant as having a history of kidney stones. We have added the relevant content to the methodology section of the manuscript.

In the context of Mendelian randomization, the cohort of individuals with urolithiasis was derived from the Finnish database, with genetic instrumental variables established based on samples exhibiting urolithiasis phenotypes. In the FinnGen consortium, 5347 cases of kidney stones were diagnosed based on N20 in the International Classification of Diseases Tenth Edition (ICD-10) and self-reported surgical codes. The accuracy of these codes for defining a kidney stone has been previously validated[2].

3. Biggest limitation is no known renal function - no creatinine values? which would be the biggest confounder. This is well established that renal stone will cause CKD and CKD is a major risk for cardiovascular disease. is it possible to obtain lab values?

Reply: We express our gratitude for the reviewer's insightful suggestions.

In response, we have incorporated the creatinine values of the patients and employed the Chronic Kidney Disease Epidemiology Collaboration (CKD-EPI) equation to calculate the estimated glomerular filtration rate (eGFR) based on serum creatinine (Scr) levels. We excluded participants with baseline eGFR values below 15 ml/min/1.73 m², as well as those who were undergoing dialysis or had received a kidney transplant at baseline. For subsequent analyses, we included participants with eGFR values categorized as ≥60, 45-59.9, 30-44.9, and 15-29.9 ml/min/1.73m². After accounting for the confounding variable of renal creatinine levels, our analysis reveals a persistent significant association between kidney stone disease and myocardial infarction, as detailed in the results section.

Reviewer #2:

Comments to the Author

1.Given the topic of kidney stone data and evaluating associations, the data is presented in an extremely complex way. For an average reader of this article, it has dense amount of statistical analysis without clarity.

Reply: We express our gratitude for the insightful feedback offered by the reviewers. In response, we have conducted a thorough reanalysis of the manuscript's statistical components, ensuring that the presentation is as clear and comprehensible as possible. Additionally, we have incorporated several images and tables into the revised manuscript to enhance its clarity and accessibility.

2. Table 1 also needs better explanation. Text states that 7377 individuals selected but N was 7282 (including the Forest plot).

Reply: We gratefully appreciate for your valuable suggestion.

Firstly, the description of 7377 individuals in the text is incorrect and the original text should be revised to “In our observational research, we selected 7,282 individuals”. Table 1 illustrates the baseline characteristics of participants by kidney stone disease status. Following the re-conduct of the statistical analysis, the sample size (n) should be modified to 7210. We have conducted a reanalysis and detailed corrections in the manuscript.

3. Secondly, what is the "N" in No or Yes? That has no explanation and these multiple "N" categories are confusing.

Reply: We extend our sincere appreciation for the reviewer's insightful suggestions.

The NHANES database employs a complex, multi-stage probability sampling design, resulting in unequal probabilities of individual selection. Consequently, it is necessary to adjust for these unequal probabilities in subsequent analyses to ensure that the individual random sampling aligns with the overall distribution of sample characteristics.

In Table 1 of the original text, the "N" associated with the "No" or "Yes" responses denote the weighted population size, which differs from the actual sample size, n, that is unweighted. Typically, N=weight * n, and the proportions derived from N more accurately reflect the distribution of the actual population in the United States. In Table 1, n represents the number of unweighted observed samples, while the percentage (%) reflects the weighted proportion. To mitigate potential confusion arising from varied terminologies, we have opted to eliminate the weighted sample size N and consistently represent the population size using the unweighted sample size n. But subsequent analysis is based on the situation that can reflect the actual population distribution.

4. In section 3.1.2 authors state heart failure was not significant but In Table 2, Heart Failure appears to be statistically significant. There seems to be discrepancy.

Reply: Thank you for the valuable feedback provided by the reviewer. It may be that I did not clearly state the OR meaning of each variable.

Table 2 in the original manuscript is a multivariate model controlling for age, sex, race, education attainment, BMI category, CRP status, diabetes, hypertension, heart failure, gout, hypertension, hypercholesterolemia, smoked 100 cigarettes, total calories, calcium intake. Therefore, the meaning of heart failure OR here should be that patients with heart failure have a higher adjusted OR for AMI (OR, 9.15 [95% CI, 5.81-15.5]).

To address potential misunderstandings associated with the original Table 2, we have revised it and presented an updated version. In this new Table 2, we have adjusted four models. Model 1 includes no variable adjustments, whereas Model 2 accounts for demographic characteristics such as age, gender, race, and educational attainment. Model 3 incorporates additional adjustments for BMI category, history of smoking (defined as having smoked 100 cigarettes), CRP status, eGFR, diabetes, gout, hypercholesterolemia, total caloric intake and calcium intake. Furthermore, Model 4 includes adjustments for other cardiovascular diseases beyond the primary outcome. The results indicate that the association between kidney stones and myocardial infarction remains statistically significant across all four models. For a detailed result, please refer to Section 3.1.2 of the article, which addresses the relationship between kidney stone disease (KSD) and cardiovascular events.

Reviewer #3:

Comments to the Author

1. The manuscript explores a relevant topic with valuable implications and is based on a sound methodological framework. However, improvements are needed in clarity, organization, data presentation, and language to strengthen its overall quality and impact.

Reply: We express our sincere gratitude to the reviewers for their constructive feedback aimed at improving the clarity, organization, data presentation, and linguistic quality of our manuscript. These invaluable suggestions will undoubtedly enhance the rigor and accessibility of our research.

In terms of clarity, we have removed seemingly plausible conclusions, made further modifications to the results section, and redrawn unclear charts. For details, please refer to the results section of the manuscript.

In terms of organization, the introduction and discussion sections have been restructured, and the logic and causal inference sections have been further revised and integrated. For specific details, please consult the manuscript.

In terms of data presentation, we have added a process analysis on Mendelian randomization. This will facilitate a clearer comprehension of the Mendelian randomization process and its associated implications for readers (Fig 1, Fig 2 in the revised manuscript). We have revised the forest plot illustrating the interactions within the subgroup analysis to enhance the clarity and immediate comprehensibility of the data (Fig 2, S1 Fig in the revised manuscript). Besides, to enhance the rigor of the Mendelian randomization process, we have incorporated pertinent scatter plots and forest plots into the supplementary materials (Fig 3, S2 Fig in the revised manuscript and Fig 4, S3 Fig in the revised manuscript)).

In terms of language, we apologize for the poor language of our manuscript. We worked on the manuscript for a long time and the repeated addition and removal of sentences and sections obviously led to poor readability. We have now worked on both language and readability and have also involved native English speakers for language corrections. We really hope that the flow and language level have been substantially improved. we have polished our manuscript carefully and corrected the grammatical, styling, and typos found in our manuscript.

Major comments

1. The objectives of the study need to be stated more clearly in the introduction. Currently, the aims are somewhat vague and scattered.

Reply: We express our gratitude for the reviewer's insightful suggestions. In response, we have restructured the introduction and incorporated a distinct paragraph to elucidate the purpose of our research. The details are as follows: “… Studies have demonstrated that the development of nephrolithiasis is affected by genetic predispositions, environmental conditions, dietary habits, physical activity levels, and various other determinants. Kidney stones are linked to systemic conditions, including hypertension, diabetes, and dyslipidemia, which are recognized as risk factors for cardiovascular disease (CVD). Consequently, it is imperative to elucidate the relationship between kidney stones and cardiovascular disease. … Given the rising incidence of kidney stones over the years, a precise evaluation of their associated cardiovascular risks is essential for alerting patients and facilitating effective health interventions. Patients with kidney stones should focus on modifying cardiovascular risk factors to mitigate the comorbidity risk of both conditions. Furthermore, elucidating the relationship between KSD and cardiovascular events is of substantial importance for advancing our understanding of the etiology of these diseases. We conducted a cross-sectional study based on NHANES to determine the relationship between kidney stones and cardiovascular events. Following this, a two-sample Mendelian randomization (MR) was applied to further explore the causal relationship between these two conditions, employing genetic markers associated with kidney stones as instrumental variables to infer causal relationships with specific cardiovascular disease outcomes. This approach is more resistant to biases associated with confounding variables and the issue of reverse causality that can plague traditional observational research” (Page 9-10 in the revised manuscript).

2. More details should be provided regarding the study design and participant selection. For instance, information about sample size calculation, inclusion/exclusion criteria,

Reply: Thank you very much indeed for your comments.

Owing to the cross-sectional design of the study and accompanying literature review, the global prevalence of kidney stones is estimated to be approximately 7.2-7.7%. The sample size for a cross-sectional study is calculated using the formula: n=(Zσ2×pq)/d2, where Zσ represents the significance test statistic. For a significance level α of 0.05, Zσ is 1.96. Here, p is the estimated prevalence of kidney stones, set at 8%, q=1-p, and d is the allowable error, specified as 0.02. Using these parameters, the minimum required sample size n is calculated to be 707. Accounting for a potential data and quality loss of 10%, the adjusted minimum sample size required is 785. Based on the established inclusion and exclusion criteria, a total of 7210 participants were included in the study, thereby exceeding the minimum sample size requirement. In my study, logistic regression analysis was used with 16 variables. According to previous research, the sample size should be 5, 10, or 20 times the number of variables[3]. A sample size of 320 would comply with the strictest empirical rule. The inclusion of 7210 research subjects also meets the sample size requirements for multivariate regression analysis.

Due to the lack of relevant content on sample size calculation in the methodology section of most literature, our description of sample size is as follows: “This study selected two NHANES cycles from 2007 to 2010 to assess the association between kidney stones and cardiovascular events. A total of 20,688 participants were initially enrolled. After excluding participants aged < 18 years (n = 7933), those with incomplete data on KSD (n = 645) or cardiovascular events (n = 61), those with incomplete data on other covariates (n = 4767), and those with recorded estimated glomerular filtration rate (eGFR) values under 15 ml/min/1.73 m², as well as those who had undergone dialysis or kidney transplantation (n = 72). 7210 participants were included in our final analysis”.

In a word, our inclusion criteria include: This study selected two NHANES cycles from 2007 to 2010 to assess the association between kidney stones and cardiovascular events. A total of 20688 participants were initially enrolled. Exclusion criteria include: (A) participants aged <18 years(n = 7933), (B) participants with incomplete data on kidney stone disease (n = 645) or cardiovascular events (n = 61), (C) participants with incomplete data on other covariates (n = 4767), and participants with recorded estimated glomerular filtration rate (eGFR) values under 15 ml/min/1.73 m², as well as those who had undergone dialysis or kidney transplantation (n = 72).

3. Tables and figures need to be formatted more clearly, with consistent use of labels, units, and statistical indicators (e.g., p-values, confidence intervals). Some of the figures are difficult to interpret due to lack of descriptive legends and clarity.

Reply: We agree with this comment and have updated Table 1. We have implemented the relevant modifications to the description in the revised manuscript: “In our observational study, we selected 7,210 individuals. Table 1 presents the baseline characteristics of the participants according to KSD status. We compared the demographic differences between participants with and without kidney stones (Table 1). Participants with kidney stones were significantly more likely to be older (55.7 versus 50.6 years), have a higher incidence of diabetes mellitus (17.5% versus 9.3%), a higher risk of gout (9.8% versus 4.1%), and a higher risk of hypercholesterolemia (51.3% versus 41.1%) (all P < 0.001).”

Table 2 will be revised and updated for presentation purposes, ensuring the use of consistent labels, units, and statistical indicators. Additionally, descriptive explanations for each model will be incorporated. The details are as follows: “We investigated the ass

---

## [Decision Letter · Decision Letter 1]

28 Jul 2025

Kidney stone disease increases the risk of cardiovascular events

PONE-D-25-10263R1

Dear Dr. XueWen Liao,

We’re pleased to inform you that your manuscript has been judged scientifically suitable for publication and will be formally accepted for publication once it meets all outstanding technical requirements.

Kind regards,

Ricardas Radisauskas

Academic Editor

PLOS ONE

Additional Editor Comments (optional):

Reviewers' comments:

Reviewer's Responses to Questions

**Comments to the Author**

Reviewer #1: All comments have been addressed

Reviewer #3: All comments have been addressed

2. Is the manuscript technically sound, and do the data support the conclusions?

Reviewer #1: Yes

Reviewer #3: Yes

3. Has the statistical analysis been performed appropriately and rigorously?

Reviewer #1: Yes

Reviewer #3: Yes

4. Have the authors made all data underlying the findings in their manuscript fully available?

Reviewer #1: Yes

Reviewer #3: Yes

5. Is the manuscript presented in an intelligible fashion and written in standard English?

Reviewer #1: Yes

Reviewer #3: Yes

Reviewer #1: (No Response)

Reviewer #3: The manuscript presents original and well-validated findings that make a meaningful contribution to the understanding of kidney stone disease and its cardiovascular implications. With minor clarifications and edits, especially in the presentation of results and discussion of mendelian randomization interpretations, it is suitable for publication in PLOS ONE.

**Do you want your identity to be public for this peer review?** For information about this choice, including consent withdrawal, please see our Privacy Policy

Reviewer #1: No

Reviewer #3: No

---

## [Editor Report · Acceptance letter]

PONE-D-25-10263R1

PLOS ONE

Dear Dr. Liao,

I'm pleased to inform you that your manuscript has been deemed suitable for publication in PLOS ONE. Congratulations! Your manuscript is now being handed over to our production team.

Kind regards,

on behalf of

Professor Ricardas Radisauskas

Academic Editor

PLOS ONE